## SCIENCE FORUM

# Best practices to promote rigor and reproducibility in the era of sex-inclusive research

**Abstract** To enhance inclusivity and rigor, many funding agencies and journals now mandate the inclusion of females as well as males in biomedical studies. These mandates have enhanced generalizability and created unprecedented opportunities to discover sex differences. Education in sound methods to consider sex as a subgroup category has lagged behind, however, resulting in a problematic literature in which study designs, analyses, and interpretations of results are often flawed. Here, we outline best practices for complying with sex-inclusive mandates, both for studies in which sex differences are a primary focus and for those in which they are not. Our recommendations are organized within the "4 Cs of Studying Sex to Strengthen Science: Consideration, Collection, Characterization and Communication," a framework developed by the Office of Research on Women's Health at the National Institutes of Health in the United States. Following these guidelines should help researchers include females and males in their studies while at the same time upholding high standards of rigor.

**JANET W RICH-EDWARDS, DONNA L MANEY\***

## Introduction

Effective medical treatments and public health rely on rigorous, inclusive research that takes into account variation across and within populations. In response to an historical exclusion of women from biomedical studies and trials, the US government has mandated the inclusion of women in clinical investigations since 1993 (*Epstein, 2007*). Similarly, researchers funded by the National Institutes of Health (NIH) in the US have been required to include females and males in preclinical non-human animal studies since 2016 (*Clayton and Collins, 2014*). Similar sex-inclusive policies have been implemented in Canada and Europe, and many journals now require the inclusion of females and males unless otherwise justifiable (*Health Canada, 2023*; *Heidari et al., 2016*; *NIH, 2015a*; *White et al., 2021*).

The rationale for such policies is founded on observations that, on average, men and women differ in the incidence and presentation of disease, risk factors for disease, treatment response and adverse drug reactions, as well as on the belief that similar, undiscovered differences may be widespread (*van Anders, 2022*; *Clayton, 2018*;

*Karp et al., 2017*). Indeed, differences between men and women have been reported in the incidence and average age of onset of many diseases, notably auto-immune diseases such as lupus and multiple sclerosis (*Voskuhl, 2011*), mental health conditions such as depression (*SAMHSA, 2012*) and schizophrenia (*Abel et al., 2010*), substance use disorders (*McHugh et al., 2018*; *Becker and Koob, 2016*), and cardiovascular disease (*Oneglia et al., 2020*; *Mosca et al., 2011*; *Ji et al., 2022*). Diabetes and cigarette smoking are more potent risk factors for cardiovascular disease among women (*Humphries et al., 2017*), and women are more likely to present with atypical symptoms of coronary heart disease than are men (*Canto et al., 2012*; *Mieres et al., 2011*; *Eastwood et al., 2013*). Observations of such differences have, in rare cases, prompted different treatment recommendations for men and women. In 2013, for example, the US Food and Drug Administration (FDA) recommended lower dosing of the prescription drug zolpidem for women than for men, citing higher blood levels of the drug in women eight hours after it was taken (*FDA, 2018*). Overall, widespread findings of statistically significant group differences

**\*For correspondence:** dmaney@emory.edu

**Competing interest:** The authors declare that no competing interests exist.

between men and women have led some to conclude that there are important differences between women and men yet to be discovered and that these differences will ultimately necessitate tailored treatments for each sex (*Clayton, 2018*).

As a result of the policies to increase inclusion, women are now represented in most clinical trials (*Bierer and Meloney, 2022*; *Sosinsky et al., 2022*; *Feldman et al., 2019*) and preclinical research increasingly includes female animals (*Beery and Zucker, 2011*; *Woitowich et al., 2020*). This change is a welcome and needed corrective; ignoring an entire sex or gender in research design, data analysis, and reporting, without strong justification, is arguably incompatible with rigorous or generalizable science. As we strive to right historical wrongs, we must maintain the high standards required of all scientific endeavors. This is easier said than done.

Sex-based research policies do not universally mandate direct comparisons between females and males. The NIH policy on sex as a biological variable (SABV), for example, explicitly states that although data must be disaggregated by sex, statistical comparisons are not required (*NIH, 2020*). Nonetheless, most researchers who comply with SABV do compare the sexes (*Garcia-Sifuentes and Maney, 2021*), and this practice appears to have led to a large increase in the number of differences reported (*Maney and Rich-Edwards, 2023*). For many researchers, including females and males represents a novel approach for which training could be lacking (*Maney et al., 2023*). Indeed, it is becoming more apparent that many claims of sex differences are based on flawed study designs, analyses or interpretations of results (*Patsopoulos et al., 2007*; *Zell et al., 2015*; *Wallach et al., 2016*; *Kaul, 2017*; *David et al., 2018*; *Eliot et al., 2021*; *Garcia-Sifuentes and Maney, 2021*; *Maney et al., 2023*; *Maney and Rich-Edwards, 2023*). The widespread use of suboptimal approaches suggests that training in methods to detect sex differences has not kept pace with the unprecedented opportunity to investigate them. Here, we seek to highlight and help address this methodological barrier by proposing best practices in study design, data analysis, and presentation of findings, emphasizing principles common to preclinical, clinical, and population science.

## Exploratory vs. confirmatory research: An important distinction

The main elaboration we propose is to remind ourselves of the fundamental difference between exploratory (hypothesis-generating) research and confirmatory (hypothesis-testing) research. In its guide to reviewers, NIH distinguishes studies "intended to test for sex differences" from those that are not, seemingly marshalling two different approaches (*NIH, 2015b*). The NIH requires only studies intended to test for sex differences to have the statistical power and analytic methods adequate to the task; others are not held to such expectations – that is, their approach to sex as a variable may be more exploratory in nature (*NIH, 2015b*; *NIH, 2020*). In differentiating exploratory studies from those designed to interrogate a sex difference, the NIH effectively proposes two standards to which NIH-funded studies can be held. This is not inherently problematic; exploratory research, by definition, operates under a different standard.

The distinction between exploratory and confirmatory research has a long history that is useful to consider in the context of recent mandates (*Forstmeier et al., 2017*; *Wagenmakers et al., 2012*). In general, exploratory studies typically include neither a priori hypotheses nor sample sizes large enough to test for effects within or between subgroups (*Schwab and Held, 2020*). The strength of exploratory studies is that they allow for and acknowledge unexpected findings that can in turn be used to generate novel hypotheses; whether such findings are happy or unfortunate accidents must be left to future studies. In contrast, authors of confirmatory studies are motivated by preliminary data or prior literature to specify clear testable hypotheses, prespecify subgroup contrasts, and size their studies to formally test for subgroup differences. Exploratory research fertilizes the farm; confirmatory research separates the wheat from the weeds. By embracing both exploratory and confirmatory research – and not muddying the distinction – we can reap the strengths of both as we seek to make our work generalizable and inclusive.

The trouble comes in the expectation that within-sex analyses should be conducted in all studies – even those not adequately powered or designed to examine how treatment effects vary by sex (*NIH, 2020*; *CIHR, 2023*). As we explain below, an underpowered, within-subgroup analysis does not meet basic standards of analytical rigor, even when taking an exploratory approach

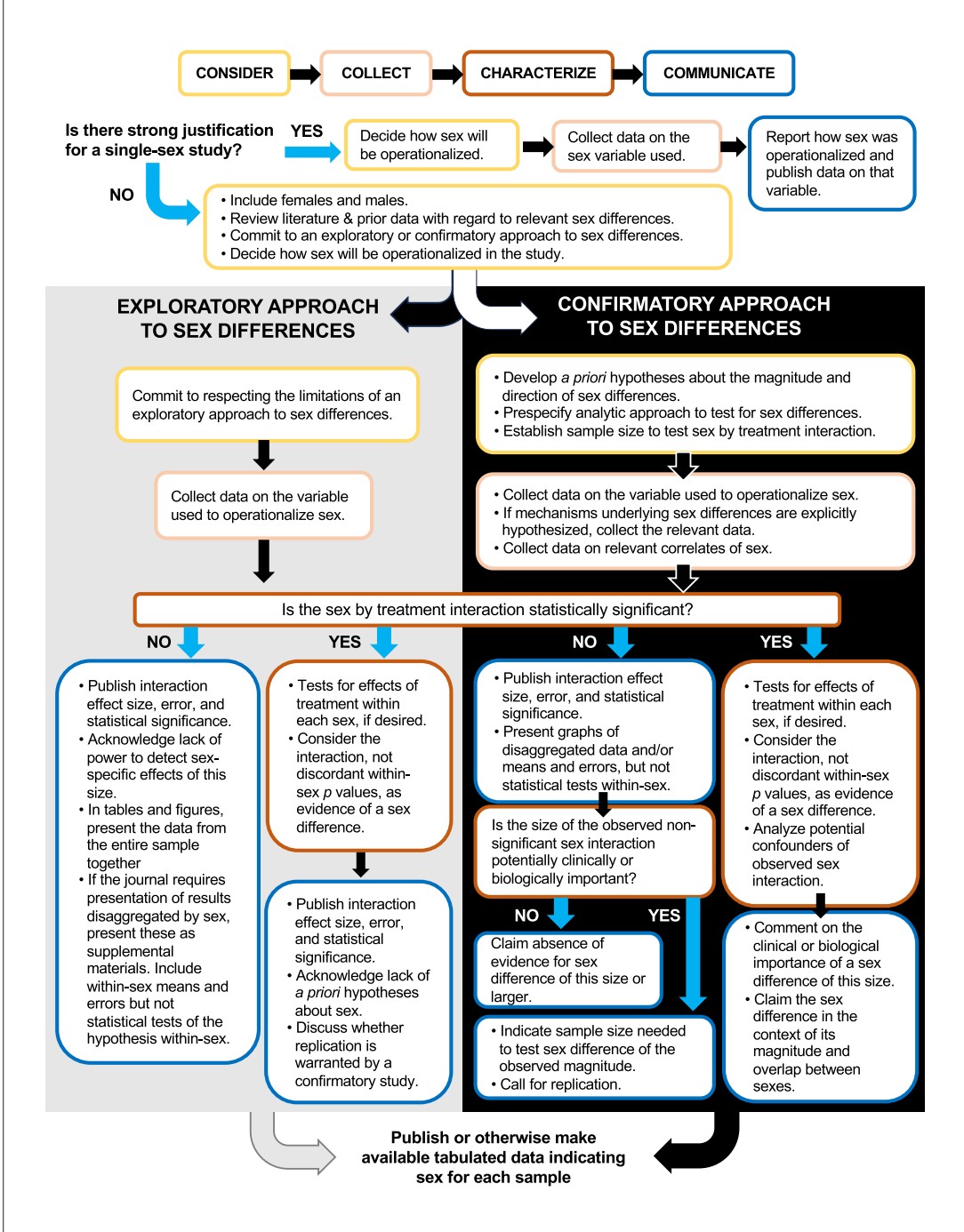

**Figure 1.** Exploratory and confirmatory research in the "4 Cs" framework for studying sex. Decision tree depicting how the "4 Cs" framework can be applied to either an exploratory or confirmatory approach to testing for sex differences. Best practices are shown for testing for a sex difference in a treatment effect (or exposure-outcome association). For observational studies, "exposure" should be substituted for "treatment".

to sex differences, and therefore threatens reproducibility (***Rich Edwards et al., 2018***). Furthermore, studies not designed to interrogate sex differences are often poorly equipped to test alternative explanations for sex effects, such as confounding by gender (***Ritz and Greaves,*** ***2022***). Finally, while an explosion of exploratory analyses will undoubtedly promote discovery of sex differences, they will also maximize false positive discoveries (***Maney and Rich-Edwards,*** ***2023***).

Below, we offer some tips to researchers who strive to comply with new guidelines while maintaining standards of rigor. Our approach is based on the "4 Cs" framework for studying sex, developed by the Office of Research on Women's Health at the NIH (https://orwh.od.nih.gov/sex-gender/nih-policy-sex-biological-variable). This framework calls upon researchers to: **Consider** sex in study design; **Collect** sex-based data; **Characterize** sex-based data in analysis; and **Communicate** sex-based data through publication. We will elaborate upon and critique elements of the 4 Cs framework, naming ongoing challenges presented by SABV and similar policies, and proposing what we hope are practical solutions. Our recommendations are outlined in *Figure 1*, which depicts largely separate pathways for exploratory and confirmatory approaches to sex differences.

Two prefatory notes before we continue. First, the principles outlined here apply to observational studies as well as experiments and clinical trials, and apply equally to pre-clinical, clinical and population science. Throughout, where we use "treatment" or "treatment effect", observational researchers can substitute "exposure" or "exposure-outcome association" respectively. Second, we note that gender – which encompasses identity, expression, and sociocultural expectations – also affects human health and is already the subject of directives in some countries (*Health Canada, 2023*; *NASEM, 2022*). While this article highlights sex as the focus of current federal mandates in the US, the same research principles outlined below apply to the study of gender or any other subgroup.

## Consideration phase: Consider sex *and* commit to an exploratory or confirmatory approach to sex differences

In this phase of study design, investigators should think about the potential impact of sex on the phenomenon under study and decide whether their hypotheses will include sex differences. This step includes consideration of the relevant literature to identify evidence for plausible effects of sex or sex-related factors on the variables of interest. It is at this stage, early in the research process, that researchers commit to either an exploratory or confirmatory approach to sex differences.

We recommend the following steps for the **Consideration** phase of all investigations (*Figure 1*):

- Explicitly operationalize sex. The NIH defines sex as "a multidimensional construct based on a cluster of anatomical and physiological traits that include external genitalia, secondary sex characteristics, gonads, chromosomes, and hormones" (*Barr and Temkin, 2022*). Taken individually, none of the traits in the cluster can by themselves define a body as "male" or "female." Thus, the term "sex" has unstable meaning (*Richardson, 2022*; *Bhargava et al., 2021*; *Massa et al., 2023*). For any particular study, the definition of sex can depend on the nature and goals of that study; in preclinical studies, for example, the sex of the research animals might be defined by anogenital distance or genotype. In most clinical and population studies, sex is defined by participant self-report (although best practices for the use of categories and checkboxes are without consensus and are rapidly changing; *NASEM, 2022*; *Suen et al., 2020*; *Kronk et al., 2022*; *Garrett-Walker and Montagno, 2023*). Depending on study aims, variables such as pubertal or menopausal status may be more useful than binary sex. Regardless, within a study, the definition should remain consistent throughout each phase, including the interpretation of results. Note that sex should be clearly operationalized even for single-sex studies.
- Unless otherwise justified, include more than one sex (e.g., females and males) to improve generalizability from a sample to the general population.
- Although sex inclusion policies typically endorse a binary approach to sex as a variable, consider whether sex must be binarized for the particular study at hand. For example, some sex-related variables that might be used to operationalize sex (e.g., hormone levels, anogenital distance, some sex-associated behaviors) are not categorical. Other authors have provided excellent additional guidance for authors seeking to avoid binary operationalizations or to take more nuanced approaches (*van Anders, 2022*; *Ritz and Greaves, 2022*; *Richardson, 2022*; *Joel and Fausto-Sterling, 2016*; *Hyde et al., 2019*; *Massa et al., 2023*).

In addition to the above, studies taking a **confirmatory** approach to sex differences require researchers to:

- State and justify hypotheses about sex differences a priori.

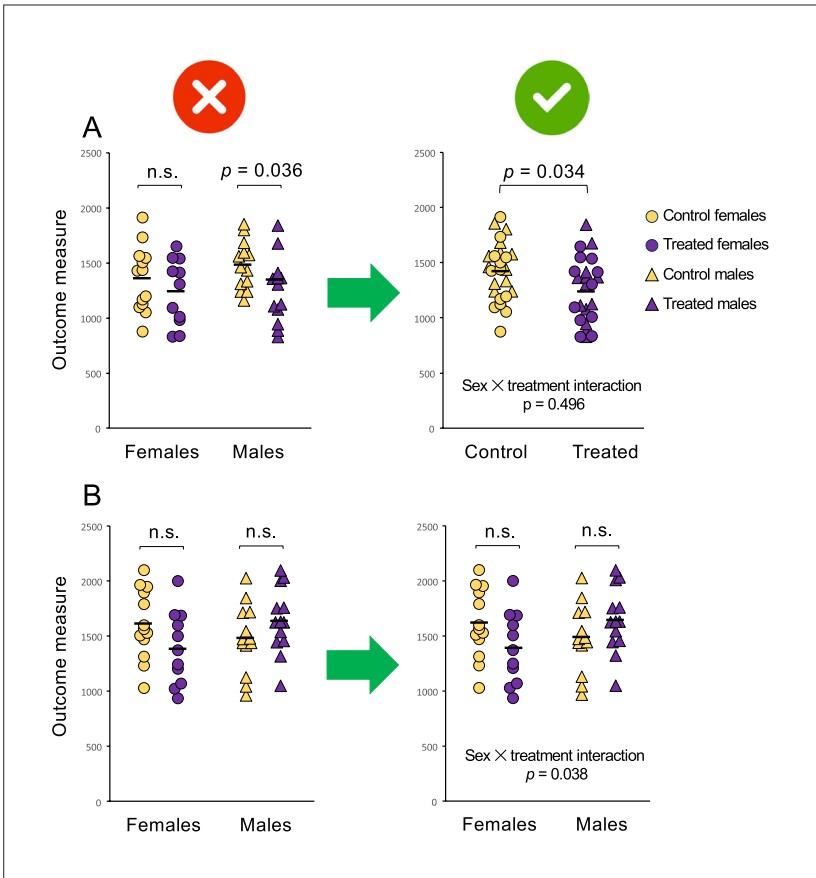

**Figure 2.** Analysis of data separately by sex can lead to erroneous conclusions. The graphs above show the response to a control (yellow) or experimental (purple) treatment in females (circles) and males (triangles). Solid black lines represent the mean within each group. The top two graphs (**A**) depict two different approaches to the same dataset, in which there is an effect of treatment irrespective of sex and no sex difference in the response to treatment. The left panel shows how analyzing the data separately by sex, that is, making the difference in sex-specific significance (DISS) error (*Maney and Rich-Edwards, 2023*), leads to a false positive finding of a sex difference. The effect of treatment is statistically significant for males (*$P$=0.036) but not for females ($P$=0.342) but concluding a sex difference in the response to treatment constitutes an error – the sexes have not been compared statistically. To compare the responses between males and females, we can test for an interaction between sex and treatment, for example in a two-way analysis of variance (ANOVA). The sex × treatment interaction is not statistically significant ($P$=0.496), meaning there is no evidence that males and females responded differently and no justification to test for effects of treatment separately within each sex. The main effect of treatment, which is estimated from all subjects combined, is significant ($P$=0.034) suggesting that the treatment altered the outcome measure irrespective of sex. To communicate the main effect of treatment, which is the important finding in this case, the sexes are combined for presentation (right panel). The bottom two graphs (**B**) depict two approaches to a different dataset, in which there is no main effect of treatment but the sexes did respond differently. The effect of treatment is not statistically significant within females ($P$=0.115) or within males ($P$=0.185), so separate analysis (DISS error) could lead to a false negative, *i.e.* concluding that the sexes responded similarly when they responded differently. The sex × treatment interaction, which is statistically significant ($P$=0.038), indicates the difference in the responses of females and males. Because of this significant interaction, the data from males and females are presented separately (right panel) and *post hoc* tests of the effect of treatment are presented for each sex. See *Figure 2—source data 1* for the data depicted in A and B.

The online version of this article includes the following source data for figure 2:

**Source data 1.** Source data for *Figure 2*, showing the response of females and males to a control or experimental treatment.

- Prespecify in the study protocol an analytic approach to test for sex differences. Most researchers conducting experiments will plan to test for sex-by-treatment interactions in a factorial design, as described below.
- Specify the expected direction and magnitude of hypothesized sex differences. Predicting the size of sex differences is essential to estimating the sample size necessary to detect them; it is often helpful to anticipate a range of magnitudes.
- Plan for a sample size that provides enough statistical power to detect sex differences of the expected size.
- As elaborated below, consider other variables that often co-vary with sex and whether steps can be taken to control for them. In human studies, for example, gendered occupations could result in different exposures; in non-human animal studies, housing only the males singly (to prevent fighting) can be similarly confounding.

Studies taking an **exploratory** approach to sex differences require neither a hypothesis about sex differences nor a sample size adequate to test for them; however, it is at this design point that an exploratory study commits itself to being transparent in reporting its exploratory nature – even if it does reveal an incidental sex difference.

## Collection phase: Collection of sex-based data

In the Consideration phase described above, we will have made a careful decision about what "sex" means in the context of our study. Sex-informed data collection, described in this section, requires equally careful consideration of what sex is *not*. The issue is one of precision. Sex is not itself a tangible, single variable; it is simply a proxy for many factors that covary imperfectly with each other (*Richardson, 2022*; *Maney, 2016*; *Massa et al., 2023*). This fact can lead to a scientific sleight-of-hand that goes unnoticed when we use sex as shorthand for a grab bag of factors correlated with sex. When possible, it is better to *measure* sex-related variables, such as hormones or gene expression, than to reason backwards from sex to speculative biologic mechanisms. When collecting such data is not possible, extra care must be taken in the Characterization and Communication phases to govern reflexive assumptions about sex and its mechanisms.

Sex is related not only to physiological variables such as gene expression but also to gendered

behaviors and social environments, which in turn affect physiology (*Ritz and Greaves, 2022*). Sex and gender thus nearly always confound each other in human studies. The effects of gender expression, gendered occupations, and disparities in access to health care can be challenging to disentangle from manifestations of 'sex,' not only because the construct of gender is complex but also because it is exquisitely sensitive to place and time (*Nielsen et al., 2021*). Even non-human animals, regardless of whether they experience something we could call gender, can experience sex-specific environmental factors such as housing density, aggression from conspecifics, access to resources, and so on *Cortes et al., 2019*. Thus, in human and non-human animal studies alike, it can be important to collect data on environmental factors that covary with and are easily conflated with sex.

*Figure 1* acknowledges requirements in the **Collection** phase for exploratory and confirmatory studies. For studies taking a **confirmatory** approach to sex differences, we recommend the following steps:

- Collect data on the variable used to operationalize sex. For example, if sex is operationalized by anogenital distance in newborn mice, measure and record that distance for each mouse.
- If the hypotheses specify potential mechanisms underlying sex differences, collect data that can provide information about those mechanisms. For example, if sex (operationalized using genotype) is hypothesized to affect the outcome variable via an association with estradiol, collect data on both genotype and estradiol. These data will lead to more informative conclusions.
- As noted above, many physiological and environmental factors covary with sex no matter how it is defined or operationalized. These factors include those that are not part of the hypothesis and therefore could potentially confound any test for sex differences, for example gendered occupational exposures or, in non-human animal studies, housing conditions. When possible, collect data on the most relevant correlates of sex (those that may affect the endpoint) to allow accounting for or adjusting for factors that could masquerade as sex.
- In clinical studies, consider allocating participants first to sex strata, then randomizing participants to treatment groups in a systematic way to balance numbers of each sex proportionally across treatments. This strategy, called sex-stratified

randomization, distributes sex and sex-related factors equally across treatment arms (*Piantadosi, 2005*). Sex-stratified randomization enhances the ability to examine interactions between treatment and sex, mitigates potential confounding bias and noise, and can increase statistical power (*Piantadosi, 2005*).

Studies taking an **exploratory** approach to sex differences need not take the above steps. For such studies, we recommend the following:

- Collect data on important potential confounders to avoid generating post hoc hypotheses about sex differences that would be better explained by other factors.
- In an exploratory approach to sex differences, sex may not be as high a priority for stratification as factors known to be strongly associated with outcome or treatment response (e.g., pre-existing health conditions, age, study site, etc.). Therefore, to maximize statistical and logistical efficiency, trialists may choose to stratify randomization on other factors. For most such studies, sex can nonetheless be included as a variable of interest in the analysis, keeping in mind that in the absence of sex-stratified randomization, within-sex analyses forfeit protection against confounding by prognostic factors that co-vary with sex (*Piantadosi, 2005*; *Cui et al., 2002*).

## Characterization phase: Analysis of sex-based data

In this section, we will address issues related to testing for sex differences. First, a note about the two main categories of sex differences: traits and effects. We may be interested in sex-based variation in certain traits, such as behaviors or the incidence of disease; these contrasts involve two variables, the trait and sex. For example, the question of whether women are more likely to get autoimmune disease than men is a simple two-variable question (the extent to which sex predicts disease status).

More often, though, when we speak of a sex difference in research findings, we are referring to the impact of sex on the association between two *other* variables. For example, we may be interested in whether sex interacts with, or modifies, the relationship between treatment and disease (in an experiment) or exposure and outcome (in an observational study). In a clinical trial, researchers may examine the influence of sex on the efficacy of aspirin to prevent heart disease;

this is a three-variable problem with sex, treatment, and outcome. An observational study may query whether the association between diabetes and future cardiovascular disease differs between men and women, a question involving sex, exposure (diabetes), and outcome. It is these three-variable questions that researchers typically face when, to comply with a new policy, they add another sex to their study. For these, comparing the sexes means testing whether the effect is stronger, or perhaps goes in a different direction, in one sex vs. another. These are important questions to answer as accurately as possible; when translated, sex interactions can influence the treatments and doses that are offered to women and men (*FDA, 2018*; *Zhao et al., 2023*).

Unfortunately, many studies fumble the three-variable question (sex/treatment/outcome) by turning it into a *pair* of two-variable problems (treatment/outcome for males and treatment/outcome for females); this practice of qualitatively comparing the within-sex p-values fails to actually contrast the sexes. For example, if we observe a statistically significant treatment effect among males but not females (or *vice versa*; left panel of *Figure 2A*), we may prematurely claim a "sex difference" or a "sex-specific effect." Similarly, we might mistakenly declare the response to treatment as the "same" if the within-sex tests were either both significant or both not significant (left panel of *Figure 2B*). In fact, we have not directly compared the treatment effect between the sexes at all. We have instead committed a logical error so widespread among subgroup analyses that it has been called out as one of the most common statistical mistakes in science (*Makin and Orban de Xivry, 2019*; *Allison et al., 2016*). The error is easy to commit (and both authors of this article have unwittingly made it in their earlier work). In the context of sex differences, where it is rampant, we have proposed calling it the difference in sex-specific significance (DISS) error (*Maney and Rich-Edwards, 2023*).

DISS is an error for several reasons. First and foremost, within-group analyses tell us nothing about between-group differences; when we perform the 'parallel play' of separate analyses by sex, we have neither quantified a sex difference in treatment effect nor considered the role of chance in its appearance (*Maney and Rich-Edwards, 2023*). Second, p-values are driven by factors (such as subgroup sample size) that have no bearing on the treatment effects researchers intend to compare. Third, contrasting p-values above and below an arbitrary threshold such as 0.05 tempts researchers to categorically declare

an effect in one sex and a lack of effect in the other, forgetting that a statistical test can be used only to reject a null hypothesis, not support one (*Maney, 2015*).

To provide real evidence of sex differences in the response to treatment, we must *directly* compare the treatment effects between males and females (*Nieuwenhuis et al., 2011*). In the biomedical sciences, this test usually consists of an interaction term (sex-by-treatment) that is included in the statistical model, along with the main effects of the factors sex and treatment. In preclinical research, if the main effects in the model are categorical, the design is typically called 'factorial,' meaning there are multiple factors of interest, such as sex and treatment. In clinical and population science, the main effect of sex and its interactions with treatment are typically captured by terms included in a regression model. The principle, however, is the same no matter the method: the sex-by-treatment interaction term represents the difference in treatment effect between males and females. If this interaction is statistically significant, only then do we have evidence that the sexes responded differently to the treatment. In that case, *post hoc* tests of the treatment effect could be done within each sex but the resulting sex-specific p-values would not provide more information about the size or statistical significance of the sex difference itself.

Despite the invalidity of the DISS approach, it is widespread in sex differences research (*Garcia-Sifuentes and Maney, 2021*). Its adoption may be attributable to several factors. First, statistical interactions can require notoriously large sample sizes to detect (*Galea et al., 2020*); authors of underpowered exploratory studies sometimes decide to skip testing for them altogether and resort instead to a DISS approach. But underpowered studies are particularly vulnerable to DISS errors; as power decreases, the odds of discordant p values in the female and male subgroups increase rapidly. Consider, for example, an experiment with equal numbers of males and females designed to have 80% statistical power to detect a non-null ($P<0.05$) treatment effect in the entire sample. Such a study typically has only 50% power to detect a within-sex treatment effect. Under this common scenario, when the treatment effect is non-null and does not, in fact, vary by sex, the likelihood of one group yielding $P<0.05$ and the other $P>0.05$ is 50% (*Bland and Altman, 2011*; *George et al., 2016*; *Brookes et al., 2004*). In other words, half the time, by chance alone, the treatment effect within males will be statistically significant while that for females is not, or *vice versa*. One might as well flip a coin to declare a sex difference – obviously not a strategy that promotes rigor and reproducibility. A rigorous strategy would be to test for a sex-by-treatment interaction regardless of whether we are powered to detect one and to accept the limitations of our exploratory approach.

Second, perhaps inspired by calls to find sex differences (*Shansky and Murphy, 2021*; *Rechlin et al., 2021*; *Tannenbaum et al., 2019*), authors may be concerned about missing them and therefore choose a less conservative approach. But, although the DISS approach is biased toward false positive findings (*Bland and Altman, 2011*; *George et al., 2016*), it can also produce false negatives – that is, it can cause investigators to miss real sex differences (left panel of *Figure 2B*). For example, in a recent study of abdominal obesity in children (AO), the authors missed a large sex difference in the association between AO and a measure of lipoprotein particle number because their within-sex p-values showed non-significant associations in both girls and boys (*Akiyama et al., 2022*). In fact, the interaction between sex and AO was highly significant ($P=0.001$); the association in girls was positive and among boys, negative (*Vorland et al., 2023*). The significant interaction serves as strong evidence that the association between AO and this measure of lipoprotein *does* depend on sex – a potentially important finding that was masked by a DISS approach.

Finally, some investigators may adopt a DISS approach because they believe it accounts for baseline sex variability (i.e., noise) that could mask effects of the other variables of interest. They could also be concerned that effects of treatment could vary so much between the sexes that a main effect is cancelled out. But in the case of a baseline sex difference, the better strategy is still to include sex as a variable of interest; the main effect of sex captures any baseline variation due to sex (*Phillips et al., 2023*). Including sex in the statistical model can have the added advantage of unmasking effects of treatment when the effect is much larger in one sex than another or goes in different directions in each sex (*Phillips et al., 2023*; *Buch et al., 2019*). Similarly, clinical and population researchers can include sex as a covariate in their regression models to reduce extraneous variation and control for confounding by sex; this practice is particularly important in observational studies or in randomized trials that were not stratified by sex.

We should point out here that the main effect of sex is often confused with the sex-by-treatment

interaction term; that is, a significant main effect of sex is misinterpreted as a sex difference in the response to treatment (*Eliot et al., 2023*). In the example above regarding the effect of aspirin on heart disease, a significant sex-by-treatment interaction would provide evidence that the effect of aspirin on heart disease differed between men and women; in contrast, a significant main effect of sex simply means that there is a sex difference in the incidence of heart disease. It does not indicate a sex difference in the efficacy of aspirin. This mistaken interpretation of the outputs of analyses of variance (ANOVAs) is common (*Garcia-Sifuentes and Maney, 2021*) and, notably, has been endorsed even in NIH training materials about how to incorporate sex as a variable into research designs (*NIH, 2020*).

Our recommendations for the **Characterization** (analysis) phase are provided in *Figure 1*. For all studies, our recommendations are the following:

- Test for a statistical interaction between sex and treatment. This test should be conducted regardless of statistical power.
- Note the statistical significance and magnitude (effect size) of the interaction and avoid the trap of conflating the p-value and the effect size. For example, it is possible for a large sex difference to fall short of statistical significance in a very small study (and conversely, for a trivial sex difference to be highly statistically significant in a very large study). The magnitude of an interaction is captured by the eta squared from an ANOVA or the beta coefficient for the sex-by-treatment interaction term in a regression.
- Interpret a significant effect of sex (e.g., the "main effect of sex" in an ANOVA) as a sex difference that does not depend on the treatment.

If the interaction between sex and treatment is statistically significant:

- The interaction alone is sufficient evidence for a sex difference in the response to treatment.
- Be aware that *post hoc* tests for a significant effect of treatment within each sex do not provide information about the sex difference, since the practice of contrasting subgroup p-values remains an illogical DISS error even in the presence of a statistically significant interaction.
- Instead of comparing p-values, consider the magnitude (effect sizes) and directions of the within-sex effects.

If the interaction is not statistically significant, do **not** test for effects of treatment/exposure within sex.

Not all studies fall into the category of three-variable, meaning that there may be only two or more than three variables of interest. Studies focused on the effect of sex on the endpoint, for example (the two variable trait question outlined above), will not be testing the impact of sex on a treatment effect; in these cases, the statistical comparison of the sexes might be as simple as a t-test or chi-square test. Potential confounders of sex, such as gender-related exposures, should be considered and accounted for, usually by adding such potential confounders as covariates in a regression model.

For complex designs with multiple variables of interest (e.g., treatment, sex, time, knockout genotype, etc.), it is good practice to include all interaction terms in the model to examine interactions between sex and each other factor, three-way interactions with sex, etc. as appropriate for the hypotheses. Doing so is important if you wish to draw conclusions about the extent to which interactions among the other variables depended on or differed by sex. For clinical or population studies in which many variables are measured, including all interaction terms may not be feasible, however.

## Communication phase: Presenting and interpreting sex differences

Findings of sex differences are often used by non-scientists and scientists alike to shape public policy (*Maney, 2015*; *Maney, 2016*). It is therefore critical that such findings be presented in a transparent way that is neither misleading nor dogmatic. In this section, we first consider how to present sex-based data in both exploratory and confirmatory contexts. Then, we discuss pitfalls related to interpretation of data and communication of conclusions.

### Presentation of sex-based data

Above, we made the argument that statistical approaches to sex-based data should be similar for both exploratory and confirmatory studies; namely, we should test for statistical interactions between sex and other factors, such as treatment or exposure, no matter the statistical power. The steps of the Communication phase are dictated both by the outcome of that statistical interaction test and whether the study is taking

an exploratory or confirmatory approach to sex differences (**Figure 1**).

In any study, the presentation of data is informed by the hypotheses. The figures and tables highlight the planned comparisons and depict the extent to which the a priori predictions were borne out in the data. Exploratory studies typically begin without a priori hypotheses and predictions about sex; thus there is little reason for a reader to expect results to be presented separately by sex, particularly if the findings regarding sex differences are null. Whether or not a sex-by-treatment interaction is statistically significant, it is critical to report both the magnitude (effect size) of the interaction as well as its standard error, p-value and/or confidence interval. Publishing these statistics will help future researchers decide whether to test formally for sex differences, and if they do, to calculate the sample size required to detect them.

Our recommendation for presenting findings of **exploratory** studies are as follows:

If the sex-by-treatment interaction is *not* statistically significant:

- Present means, errors, and graphs for the entire sample (not separated out by sex) in the main body of the paper. If the data are being graphed in a way that shows individual data points (e.g., scatterplots, dotplots), it can be informative to use different symbols for males and females (see right panel of **Figure 2A**). Note that this recommendation does not mean that sex should not be included in the statistical model; on the contrary, we strongly recommend including sex as a variable of interest in all statistical models and publishing those results.
- If the journal requires results to be presented separately for females and males, do so in Supplementary Material without within-sex statistical tests of treatment effects (which beg the reader to commit a DISS error even when the authors have not done so).
- In the Discussion, acknowledge that the study was underpowered to detect sex-specific treatment effects of this size.

If the interaction *is* statistically significant:

- It is usually appropriate to present tables and graphs with data separated by sex, with the caveat that the approach was exploratory and that the sex difference needs replication.
- Occasionally, researchers may judge a statistically significant sex-by-treatment interaction found using an exploratory

approach to be not particularly important to the research question. For example, the effect could be quite small (e.g., in the case of very large sample sizes) or there could be reason to believe it is spurious (e.g., in the case of obvious potential confounders). In these rare cases, it may be more appropriate to present tables and graphs showing the data for the entire sample, not separated by sex. As noted above, supplementary material can be used to present findings for females and males separately, depending on journal policy. In addition, as stated previously, no matter the result, it is imperative to publish the magnitude of the interaction (the eta squared from an ANOVA or the beta coefficient for the interaction from a regression) in addition to the *F* and *p* values for the interactions with sex.
- Authors should acknowledge the exploratory approach and may wish to call for a larger study to confirm the finding of a sex difference.

In all cases, whenever possible, publish or make available raw data with sex indicated for each sample.

**Confirmatory** studies differ from exploratory ones in that the authors are keenly interested in sex differences—they have powered their study to detect them and have clear a priori hypotheses and predictions about how their results are expected to differ by sex. Thus, as a matter of course the data will be presented in a way that depicts the sex comparison regardless of whether a difference is found. As is the case for an exploratory study, a positive finding of a sex-specific effect in a confirmatory study is evidenced by a significant interaction between sex and another factor, not by within-sex tests. For researchers wishing to indicate the sex difference on a graph, the p-value for the sex-by-treatment interaction can be depicted either on the graph itself or in a caption (see right panels in **Figure 2**).

We caution that not all sex differences, even statistically significant ones, are large enough to be important. Big data analyses in particular often have enough statistical power to unearth truly trivial sex differences. Whether a sex difference is clinically meaningful or actionable depends very much on context and must be judged on the size of the difference, not the size of the *p*-value (**Klein et al., 2015**).

### Drawing conclusions from sex-based data
As we interpret findings of sex differences and place them into context, we should consider a

number of common pitfalls. The first is the failure to be transparent about unexpected findings of sex differences from exploratory research – including statistically significant interactions. We have all had unexpected findings jump from our data; conservative treatment requires us to indicate when these are surprises. We must avoid the temptation of HARKing (Hypothesizing After Results are Known; *Kerr, 1998*); such a posteriori justification of an unplanned result is hard for readers to detect unless study protocols are pre-registered (as is required for clinical trials). Conservative communication of exploratory findings relies upon researchers' honor; indeed, the reputation of the field rests on such transparency.

Transparency is especially important in exploratory research. If looking for a sex difference is justified neither by biology nor by prior data, the probability of chancing upon a *true* positive is lower than for confirmatory research. Kent and colleagues estimated that only one in four subgroup interactions observed in exploratory analyses are true positives (*Kent et al., 2018*). In contrast, when tests are justified by strong prior hypotheses – i.e., in confirmatory studies – subgroup interactions are more likely to be true as they start from a higher pre-test probability. In other words, for exploratory studies, even statistically significant findings of sex differences should be treated with caution and caveat.

Second, for both two-variable 'trait' questions and multiple-variable 'interaction' questions, some researchers may choose to speculate about the potential causes of the sex differences they find. In these cases, it is good practice to discuss both specific and alternative explanations. In terms of specificity, it is important not to fall into the trap of attributing all sex differences to unmeasured covariates. For example, there is a particular inaccuracy – verging on laziness – in the reflexive assumption that phenomena observed in females must be driven by estrogens. In terms of alternative explanations, observational studies must at least consider the role that gendered exposures and experiences might play; not every putative sex difference can be explained by the usual "biological" suspects (*Ritz and Greaves, 2022*).

Finally, we must be wary of sex essentialism and reification. Most human societies (including some study sections and promotions committees) can be fascinated by sex differences, no matter how small. The failure to consider large overlap between sexes can lead us to inflate the importance of sex differences and reify notions of sex that our investigations should seek to query.

The unfortunate result of an over-emphasis on sex, beyond what is warranted by the data, will be a "precision medicine" plagued by imprecision – separate treatments for women and men that do not consider individual physiologies or the myriad traits that predict treatment efficacy better than sex. At this time, at least one treatment has been made less available to women in the US on the basis of "sex differences" that are relatively small and of controversial importance (*Zhao et al., 2023*). Availability of other treatments and preventatives are in danger of being similarly curtailed for women without sufficient evidence (*Denly, 2021*; *Tadount et al., 2020*). It is incumbent upon researchers to consider our own biases and to anticipate the ways in which our findings will be used by clinicians and spun by the media.

Our recommendations for reporting interpretations of sex comparisons are as follows:

- Indicate whether tests for sex differences were exploratory or confirmatory.
- Refrain from manufacturing 'prior' hypotheses about sex differences after the results are known.
- Contextualize the magnitude of any sex difference and, where relevant, comment on its clinical or biological importance (or lack thereof).
- Discuss the degree of overlap in traits (or treatment effects, when possible) across sex.
- Avoid over-interpreting findings of sex difference in written and oral presentations, including media interviews.

## Conclusion

Advocates of sex-inclusive research policies have inspired a large and growing community of scientists eager to correct decades of neglect of women and females in biomedical research. The success of this movement relies on the rigor and quality of our science. In mandating sex-inclusive research without requiring sound methodology, we are attempting to reap the benefits of exploratory research without the discipline of confirmatory research. In no other arena has one *particular* subgroup analysis been singled out as appropriate for *every* study, whether or not the investigators hypothesized differences to begin with. We will find ourselves shouting sex differences from every mountaintop, simply because we searched for them under every rock. Thus, although the recent dramatic increase in reports of sex differences (*Maney and Rich-Edwards, 2023*) is encouraging, it is at the same time

concerning. While some of these discoveries will prove to be real and in fact meaningful, many will not. We advocate a return to the fundamental principles of exploratory and confirmatory research to promote rigorous, inclusive science. By adhering to best practices, including appropriate presentation and sharing of data, the research community will be better equipped to move quickly from exploratory to confirmatory studies without incurring the unnecessary and dangerous costs of false positives.

The science of sex differences has an interesting future that will almost certainly move us away from sex as a binary category. As a variable in scientific research, "sex" is problematic, an imperfect proxy for myriad traits that can covary tightly with each other (but often do not). Many authors have called for new ways of operationalizing and conceptualizing sex, not as a way to deny or minimize sex differences but rather to increase precision in our science (*Ritz and Greaves, 2022*; *Richardson, 2022*; *Joel and Fausto-Sterling, 2016*; *Hyde et al., 2019*; *Massa et al., 2023*). As we move into that future, the statistical treatment of sex as a factor will also evolve. Nonetheless, the difference between exploratory and confirmatory studies will remain, as will our conviction that claims of sex differences should be grounded in solid evidence.

## Acknowledgements

The authors thank Mara Hampson for assistance preparing the manuscript for submission, and Andrew Brown and Colby Vorland for fruitful discussions.

**Janet W Rich-Edwards** is in the Division of Women's Health, Department of Medicine, Brigham and Women's Hospital and Harvard Medical School, Boston, United States and the Department of Epidemiology, Harvard TH Chan School of Public Health, Boston, United States

 https://orcid.org/0000-0001-5066-8808

Donna L Maney is in the Department of Psychology, Emory University, Atlanta and the Radcliffe Institute for Advanced Study, Harvard University, Cambridge, United States

dmaney@emory.edu

 https://orcid.org/0000-0002-1006-2358

*Author contributions:* Janet W Rich-Edwards, Conceptualization, Data curation, Formal analysis, Writing – original draft, Writing – review and editing; Donna L Maney, Conceptualization, Data curation, Formal analysis, Funding acquisition, Writing – original draft, Writing – review and editing

*Competing interests:* The authors declare that no competing interests exist.

## Funding

| Funder | Grant reference number | Author |
| --- | --- | --- |
| National Institutes of Health | U54AG062334 | Donna L Maney |
| Radcliffe Institute for Advanced Study at Harvard University | | Donna L Maney |

The funders had no role in study design, data collection and interpretation, or the decision to submit the work for publication.

## Decision letter and Author response

Decision letter https://doi.org/10.7554/eLife.90623.sa1
Author response https://doi.org/10.7554/eLife.90623.sa2

## Additional files

### Supplementary files
• MDAR checklist

### Data availability
The example dataset presented in Figure 2 is included as *Figure 2—source data 1*.

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
