## [Decision Letter]

**Decision letter after peer review:**

Thank you for submitting your article "Best Practices in the Search for Sex Differences" to *eLife* for consideration as a Feature Article. Your article has been reviewed by three peer reviewers, and the evaluation has been overseen by two members of the *eLife* Features Team (Peter Rodgers and Hazel Walker). The following individuals involved in review of your submission have agreed to reveal their identity: Stacey Ritz.

The reviewers and editors have discussed the reviews and we have drafted this decision letter to help you prepare a revised submission.

Summary:

The implementation of the NIH's sex as a biological variable (SABV) policy, the Sex and Gender Equity in Research (SAGER) guidelines, the CIHR's requirement for researchers to address sex and gender considerations, and other journal and funder policies, mean that an increasing number of biomedical researchers are incorporating sex and gender into their research. However, many of these researchers don't have a deep grounding in understanding the complexities of sex and gender, and as a result, much of this work takes a relatively shallow view of how sex and gender might influence health and biology. This article provides advice to researchers on how to implement sex as a biological variable in both exploratory and confirmatory studies. However, there are a number of points that need to be addressed to make the article suitable for publication.

Essential revisions:

1. The title needs attention. Although it has a punchy appeal in its current form, I can't help but feel that the approach of *searching for* sex differences is, itself, one of the practices we actually want people to avoid? Certainly we want researchers to study how sex-related factors influence important biological and health phenomena, but a "search for sex differences" seems to presume the outcome of a male-female comparison, or to suggest that finding a male-female difference is a a more desirable outcome than not finding a difference.

2. Explanatory vs confirmatory studies. I don't agree with some of the recommendations, and question whether the differences between explanatory and confirmatory studies are significant enough to justify a different workflow (beyond that a confirmation needs a power calculation). Specifically:

a. Exploratory – why not use a sex stratified randomisation?

b. Both sexes could be included to estimate a generalisable effect. The experiment could still be described as confirmation or explanatory in the context of the intervention effect. Alternatively, it could be confirmatory or explanatory in terms of the exploration of the intervention effect depending on sex. This needs exploring in the manuscript and the diagram needs to give clearer context to which situation they are referring. In my conversations with both the NIH and the MRC, the sex inclusion policies are not to study sex differences but rather to deliver generalisable estimates with the scenario that if the intervention effect is very depend on sex this would be detected.

c. I question the proposal "acknowledge that discordant p values …" Wouldn't it be better practice to report the interaction p value?

d. LHS of Figure 1: if the interaction term is significant, "If desired, …" do you mean if the ES looks biological interesting?

e. LHS of Figure 1: when interaction is not significant "publish aggregated results". Why not present the statistical results of the main effect which is estimated across the two sexes? Pooling is bad practice.

f. LHS of Figure 1: interaction significant – why not comment on the importance of a sex differences of this size?

g. Is it clear, that you should be operating on the LHS when your goal is a generalisable estimate?

3. The following statements are judgemental and at odds with the research for the preclinical field.

"some loss of integrity … due to investigator … disinterest".

"After all, … sex differences.. nor even a passing interest".

4. The manuscript talks about power and the challenges in detecting sex differences but doesn't reference nor consider an important paper exploring this topic (Phillips et al., 2023). This paper shows that when the intervention effect is very different (opposite direction or only in one sex) the power does pass from the main effect to the interaction term providing evidence that it studying the two sexes separately. How does this finding intersect with their arguments?

Phillips B, Haschler TN, Karp NA (2023) Statistical simulations show that scientists need not increase overall sample size by default when including both sexes in in vivo studies. PLoS Biol 21(6): e3002129. https://doi.org/10.1371/journal.pbio.3002129

5. Page 4: "Here we seek to … address some of the methodological barriers " There have been discussions on the barriers in the preclinical space (eg Karp and Reavey 2019 British J Pharmacology 176:4107-4118) that are not considered.

It wasn't clear whether it was aimed at preclinical or clinical research to me. For example – most referencing come from clinical. Being confounded by gender is only relevant to clinical. I think it needs to be clear on scope – analytical and design barriers? It doesn't cover ethical (3Rs, reproductive safety concerns), welfare or recruitment challenges.

6. The authors make many statements that should be supported by published data – even and especially when it concerns a topic of current political interest and pressure (the authors explicitly discuss these dangers themselves). An example is the statement about the "historical exclusion of women in biomedical studies". Or the requirement to include both sexes in pre-clinical studies (which implies that it was not done before, that males were mostly used – which is not supported by the literature). Another example is that "similar policies have been implemented in Canada and Europe and many journals", which is only partially supported by the chosen literature.

Carefully check all statements throughout the manuscript as to whether they need literature support.

7. The terms sex and gender should be appropriately introduced at the very beginning (see e.g., Clayton and Tannebaum 2016). They are still mixed up frequently in literature, with using gender regularly for research animals.

8. This reviewer disagrees with the statement (page 19) that when authors consider sex differences (statistically significant) not interesting, they should present aggregated data. This approach deprives the readers of making their own judgments.

9. Some specific points:

a. "As a result of the policies to increase inclusion of both sexes, women are now represented in most clinical trials; preclinical research increasingly includes female animals" – References for animal studies are missing

b. The extremely high percentage of sex differences in gene-deficient animals (see Karp 2017 Nat Comm) points towards possible sex differences in a large number of human-centric studies as well.

c. Statistically sound (or field standards) for group sizes in exploratory research are available – would not now be the time to call for increases in group sizes (exploratory) to make SABV analysis more sound?

d. In the consideration phase: Does one not start with the decision of confirmatory versus exploratory?

e. While for preclinical research the operationalization of sex is well-explained, it would be helpful to similarly give guidance on clinical/observational research.

f. A clearer description of block and/versus factorial design may benefit the reader, as well as pointing to respective literature for animal, human, and epidemiological studies to help implement it.

10. The statement at the top of p6 reads "sex is the only subgroup analysis mandated by the federal government." I have not read every piece of guidance and policy from the ORWH or NIH on the SABV policy, but I have heard a number of advocates of the policy state unequivocally that the SABV policy does not *require* researchers to compare males and females, so this may need to be given more nuance. I think it could also be very valuable for the authors to expand more here on the dangers of such an emphasis on 'sex differences' and sex-specific interventions, as many researchers engaging in this type of approach fail to see the problems that almost inevitably arise from uncritically implementing findings based on a M-F binary.

11. It could be useful for the authors to expand and offer a few more examples for their bullet point on page 8 about considering whether sex must be binarized -- this will likely be a new possibility for many readers and they could probably benefit from a little more explanation and examples.

12. I'd like to see the authors take a stronger stand in the final bullet point about becoming familiar with the literature. People doing exploratory studies should be expected to examine whether relevant findings have already been published; and people doing confirmatory studies absolutely *should* know what the literature already says about findings and possible mechanisms. They've phrased it as "a good idea" but I'd like to see them take a stronger stand.

13. The point on page 9 about sex category as a proxy, and that it is better to measure sex-related variables where possible, is an absolutely crucial point, perhaps one of the most important points in the paper. I would encourage the authors to emphasize this further, and perhaps revisit it in the conclusions or abstract, to ensure it receives appropriate attention from readers.

14. The final bullet point on page 11, it is not clear to me what "factorial (blocked) sex-stratified randomization" means -- I would suggest offering a concrete example to illustrate.

15. In the last bullet point on page 16, the authors briefly suggest that researchers should consider effect sizes and directions of within-sex effects. This is a point that I think warrants greater emphasis and discussion, particularly because researchers focussed on sex differences tend to over-interpret the importance of a difference in means, and neglect to evaluate or discuss the extent of heterogeneity within the groups and overlap between them, leading to hyper-polarized calls for sex-specific intervention that are not justified by the data. It would be valuable to have a paragraph or two discussing the importance of effect size (and measures of it) and warning against the tendency to call for sex-specific intervention whenever there is a difference in means.

---

## [Author Response]

Essential revisions:1. The title needs attention. Although it has a punchy appeal in its current form, I can't help but feel that the approach of *searching for* sex differences is, itself, one of the practices we actually want people to avoid? Certainly we want researchers to study how sex-related factors influence important biological and health phenomena, but a "search for sex differences" seems to presume the outcome of a male-female comparison, or to suggest that finding a male-female difference is a a more desirable outcome than not finding a difference.

We have changed the title to ‘Rigor and Reproducibility in the Era of Sex-inclusive Research: Best Practices for Investigators’

2. Explanatory vs confirmatory studies. I don't agree with some of the recommendations, and question whether the differences between explanatory and confirmatory studies are significant enough to justify a different workflow (beyond that a confirmation needs a power calculation). Specifically:a. Exploratory – why not use a sex stratified randomisation?

As requested below, we have added to the text a more comprehensive treatment of sex stratified randomization, including why it may not be appropriate for all exploratory studies (when variables other than sex may have more influence on the outcome and be higher priority for stratification). As randomization is germane only to experimental studies, we have removed the bullet from the figure.

b. Both sexes could be included to estimate a generalisable effect. The experiment could still be described as confirmation or explanatory in the context of the intervention effect. Alternatively, it could be confirmatory or explanatory in terms of the exploration of the intervention effect depending on sex. This needs exploring in the manuscript and the diagram needs to give clearer context to which situation they are referring.

To clarify, the distinction we make between confirmatory and exploratory research is with respect to their different intention and preparation to examine sex differences, not whether they take a confirmatory or exploratory approach to testing the main effects of treatments. We have clarified in the text and added to the Figure that the terms “confirmatory” and “exploratory” refer to the discovery of sex differences, not the main effect under study.

In my conversations with both the NIH and the MRC, the sex inclusion policies are not to study sex differences but rather to deliver generalisable estimates with the scenario that if the intervention effect is very depend on sex this would be detected.

The NIH and MRC now require that males and females be included in studies unless there is strong justification. Explicit sex comparisons are not required. Nonetheless, most researchers who include males and females end up comparing them, often using inappropriate approaches. For example, NIH has promoted within-sex analyses which leads to the DISS approach; in their online training course (https://cihr-irsc.gc.ca/e/49347.html), CIHR explicitly directs researchers to take a DISS approach to “reveal” sex differences. Our goal is to provide guidance about how to compare the sexes when that is what researchers would like to do, and how to avoid seeming to compare the sexes when in fact no comparison was undertaken. The paragraph of the introduction in which we introduce the problem we wish to address (lack of training in appropriate methods to compare the sexes) has been rewritten to better articulate these goals.

c. I question the proposal "acknowledge that discordant p values …" Wouldn't it be better practice to report the interaction p value?

We have clarified the language by changing the figure to read: “Consider the interaction, not discordant within-sex p values, as evidence of a sex difference.” The interaction should always be reported: that point is included in the next (Communicate) bubble, which reads “Publish interaction effect size, error, and statistical significance.”

d. LHS of Figure 1: if the interaction term is significant, "If desired, …" do you mean if the ES looks biological interesting?

When the interaction is significant in an exploratory study, within-sex tests are justified but not always necessary. We have now elaborated on this point in the text. Whether to explore the finding further depends on the research question, the size of the interaction, the likelihood of confounding bias, and the researcher’s goals. Most researchers will, in fact, conduct the post-hoc tests, likely because they believe they are required to do so. Our recommendation here is to clarify that they are not, and that there are cases in which it is probably OK to not conduct them.

e. LHS of Figure 1: when interaction is not significant "publish aggregated results". Why not present the statistical results of the main effect which is estimated across the two sexes? Pooling is bad practice.

We are not advocating for pooling (ignoring sex) in the statistical model. As noted in the manuscript, sex should always be included as a variable of interest in the analysis. Here, we are arguing that in the absence of evidence that the effect of treatment depended on sex, data from males and females should be presented (in graphs, etc.) together. This does not preclude or undo the inclusion of sex as a variable of interest in the analysis of the data, nor does it interfere with reporting of the main effect of treatment (which typically takes sex into account). This has been clarified in the bulleted list.

f. LHS of Figure 1: interaction significant – why not comment on the importance of a sex differences of this size?

We wish to distinguish between the communication of sex differences reported by confirmatory studies and exploratory studies. Authors of confirmatory studies are beholden to comment on the clinical or biological importance (or lack thereof) of the sex difference under investigation, moving that information one step closer to translation. Exploratory studies may chance upon a sex difference that was never hypothesized and has never before been reported. The primary responsibility in that case is to consider whether the finding warrants replication rather than to speculate about its clinical or biological significance. To require investigators to comment on translational implications of exploratory findings would be to encourage *post hoc* rationalization of sex differences, which is particularly problematic for researchers whose main interests do not lie in that area.

g. Is it clear, that you should be operating on the LHS when your goal is a generalisable estimate?

The left-hand side of the figure summarizes best practices when the study is not designed and likely underpowered to detect a sex by treatment interaction—in this case, whether the effect is generalizable is not being rigorously tested. In exploratory research, males and females have been included in numbers generally too small to test the effect of treatment separately within each sex. Typically, what is being tested (on both sides of the figure) is a main effect of a treatment or exposure, and although such an effect may not be detectable within each sex due to small sample size (left side), it can be detected when the sexes are considered together in a factorial analysis.

3. The following statements are judgemental and at odds with the research for the preclinical field."some loss of integrity … due to investigator … disinterest"."After all, … sex differences.. nor even a passing interest".

These statements have been removed and the paragraph rewritten to clarify our rationale for the piece.

4. The manuscript talks about power and the challenges in detecting sex differences but doesn't reference nor consider an important paper exploring this topic (Phillips et al., 2023). This paper shows that when the intervention effect is very different (opposite direction or only in one sex) the power does pass from the main effect to the interaction term providing evidence that it studying the two sexes separately. How does this finding intersect with their arguments?Phillips B, Haschler TN, Karp NA (2023) Statistical simulations show that scientists need not increase overall sample size by default when including both sexes in in vivo studies. PLoS Biol 21(6): e3002129. https://doi.org/10.1371/journal.pbio.3002129

We have added a brief discussion of a scenario in which a significant interaction can help researchers detect effects that may differ between the sexes. Phillips et al. (2023) is now cited.

5. Page 4: "Here we seek to … address some of the methodological barriers " There have been discussions on the barriers in the preclinical space (eg Karp and Reavey 2019 British J Pharmacology 176:4107-4118) that are not considered.It wasn't clear whether it was aimed at preclinical or clinical research to me. For example – most referencing come from clinical. Being confounded by gender is only relevant to clinical. I think it needs to be clear on scope – analytical and design barriers? It doesn't cover ethical (3Rs, reproductive safety concerns), welfare or recruitment challenges.

This sentence has been rewritten: “Here, we seek to highlight and help address this methodological barrier by proposing “best practices” in study design, data analysis, and presentation of findings, emphasizing principles common to preclinical, clinical, and population science.”

6. The authors make many statements that should be supported by published data – even and especially when it concerns a topic of current political interest and pressure (the authors explicitly discuss these dangers themselves). An example is the statement about the "historical exclusion of women in biomedical studies".

A reference has been added about the historical exclusion of women and other references have been added throughout as appropriate.

Or the requirement to include both sexes in pre-clinical studies (which implies that it was not done before, that males were mostly used – which is not supported by the literature).

A reference to the SABV policy announcement has been added, as well as literature supporting a male bias in preclinical work prior to 2016.

Another example is that "similar policies have been implemented in Canada and Europe and many journals", which is only partially supported by the chosen literature.

A reference to White et al. 2021 has been added, which covers the policies in Canada and Europe. A reference to the UK policy has also been added. The “many journals” refers to the SAGER guidelines for which Heidari, et al. is referenced.

Carefully check all statements throughout the manuscript as to whether they need literature support.

References have been added where needed.

7. The terms sex and gender should be appropriately introduced at the very beginning (see e.g., Clayton and Tannebaum 2016). They are still mixed up frequently in literature, with using gender regularly for research animals.

The definitions of sex and gender are in flux and, although we agree the terms are sometimes used incorrectly, we believe it is beyond the scope of the present paper to delve into these definitions (any that we could choose would be oversimplified and quickly outdated). At the end of the introduction, we acknowledge that like sex, gender is also critically important in human research and we point out that all of our recommendations about how sex should be handled as a variable also apply to gender, where relevant.

8. This reviewer disagrees with the statement (page 19) that when authors consider sex differences (statistically significant) not interesting, they should present aggregated data. This approach deprives the readers of making their own judgments.

We now use “entire sample” instead of “aggregated” and have clarified that our recommendation pertains only to the visual presentation of results (e.g., figures), not data analysis or presentation of statistics.

Regarding the recommendation of different graphical approaches depending on whether the sex difference is “interesting”; we have reworded this entire section to remove the word “interesting” and to emphasize that in most cases, a significant sex-by-treatment interaction warrants graphing the results from females and males separately. In some cases the author may judge a significant interaction from an exploratory approach to be irrelevant to the research question or of trivial magnitude (e.g., in the case of very large sample sizes), and may choose to put all of the results onto the same graph. This does not deprive the interested reader of making their own judgments because all of the statistics relevant to sex, including main effects, interactions, and their magnitudes, should always be presented elsewhere in the main text, and if the journal requires separate graphs, they should be included in the supplemental material.

9. Some specific points:a. "As a result of the policies to increase inclusion of both sexes, women are now represented in most clinical trials; preclinical research increasingly includes female animals" – References for animal studies are missing.

These references have been added.

b. The extremely high percentage of sex differences in gene-deficient animals (see Karp 2017 Nat Comm) points towards possible sex differences in a large number of human-centric studies as well.

A reference to Karp et al., 2017 has been added.

c. Statistically sound (or field standards) for group sizes in exploratory research are available – would not now be the time to call for increases in group sizes (exploratory) to make SABV analysis more sound?

We remain agnostic about whether researchers should or should not increase their sample sizes so that they can properly test for sex differences. One of our main goals with the manuscript is to point out that not all studies are powered to do so, which is not itself an issue unless researchers use approaches and/or draw conclusions that are not appropriate for exploratory research. That is, we do not condemn exploratory approaches – on the contrary, we hope to provide guidance about how to make exploratory research more rigorous. We also point out that (as perhaps another reviewer noted) the SABV policy does not require comparing the sexes. This point has been clarified in the manuscript.

d. In the consideration phase: Does one not start with the decision of confirmatory versus exploratory?

Yes, that is stated as the first step in the Consideration phase in the text. We have emphasized this by adding a bullet point in the Consideration phase at the top of the decision tree in Figure 1. We have added emphasis in the text that this decision is made very early in the research process. This point is also made at the end of the section on Consideration.

e. While for preclinical research the operationalization of sex is well-explained, it would be helpful to similarly give guidance on clinical/observational research.

We have added a mention of how sex is typically measured in clinical studies; as this is a controversial and fast-evolving area, we prefer to highlight the principle rather than detail recommendations. We now direct the reader to more in-depth discussions of this topic.

f. A clearer description of block and/versus factorial design may benefit the reader, as well as pointing to respective literature for animal, human, and epidemiological studies to help implement it.

We have added a description of factorial designs for pre-clinical studies and sex-stratified randomization for clinical trials. We include references for these methods.

10. The statement at the top of p6 reads "sex is the only subgroup analysis mandated by the federal government." I have not read every piece of guidance and policy from the ORWH or NIH on the SABV policy, but I have heard a number of advocates of the policy state unequivocally that the SABV policy does not *require* researchers to compare males and females, so this may need to be given more nuance.

In our manuscript we were using the term “subgroup analysis” to mean separate, within-sex analyses of disaggregated data. This practice is not the same thing as comparing males and females. The sentence has been removed to avoid confusion.

I think it could also be very valuable for the authors to expand more here on the dangers of such an emphasis on 'sex differences' and sex-specific interventions, as many researchers engaging in this type of approach fail to see the problems that almost inevitably arise from uncritically implementing findings based on a M-F binary.

We have expanded this section of the manuscript to say a bit more about the risks of such uncritical implementations of findings.

11. It could be useful for the authors to expand and offer a few more examples for their bullet point on page 8 about considering whether sex must be binarized -- this will likely be a new possibility for many readers and they could probably benefit from a little more explanation and examples.

Two additional examples are now provided. We are aware of several other papers in press and under review (by other authors) that cover this topic in detail, and do not wish to expand this part of the paper unless/until we can cite those publications. The ideas are (as the reviewer points out) new and bold and we don’t wish to take credit, only plant the seed.

12. I'd like to see the authors take a stronger stand in the final bullet point about becoming familiar with the literature. People doing exploratory studies should be expected to examine whether relevant findings have already been published; and people doing confirmatory studies absolutely *should* know what the literature already says about findings and possible mechanisms. They've phrased it as "a good idea" but I'd like to see them take a stronger stand.

The recommendation to become familiar with that literature has been moved to the top of the Consideration section and is now more clearly relevant to both confirmatory and exploratory approaches.

13. The point on page 9 about sex category as a proxy, and that it is better to measure sex-related variables where possible, is an absolutely crucial point, perhaps one of the most important points in the paper. I would encourage the authors to emphasize this further, and perhaps revisit it in the conclusions or abstract, to ensure it receives appropriate attention from readers.

This point is now emphasized a bit more in that location; note that it comes up again in the bulleted list under “Collect” and is further emphasized there (several related points underlined). The last paragraph of the manuscript also makes a similar point, and we have expanded that paragraph as well.

14. The final bullet point on page 11, it is not clear to me what "factorial (blocked) sex-stratified randomization" means -- I would suggest offering a concrete example to illustrate.

We expanded the description of sex-stratified randomization typically used in clinical research.

15. In the last bullet point on page 16, the authors briefly suggest that researchers should consider effect sizes and directions of within-sex effects. This is a point that I think warrants greater emphasis and discussion, particularly because researchers focussed on sex differences tend to over-interpret the importance of a difference in means, and neglect to evaluate or discuss the extent of heterogeneity within the groups and overlap between them, leading to hyper-polarized calls for sex-specific intervention that are not justified by the data. It would be valuable to have a paragraph or two discussing the importance of effect size (and measures of it) and warning against the tendency to call for sex-specific intervention whenever there is a difference in means.

The context for the bullet point referenced here is to present an alternative to comparing p values – instead, we recommend that researchers look at the directions of the effects and their relative sizes to make a judgment call about the extent to which the sexes “differ” instead of comparing within-sex p-values. The point about effect sizes generally, namely acknowledging overlap between the sexes, is made much more strongly in the section on interpretation, where we recommend contextualizing the magnitudes of all differences, commenting on whether they are biologically or clinically important, discussing overlap, and avoiding over-interpretations. We warn against essentialism and reification of sex and the tendency to inflate the importance of difference. In response to this comment, we have expanded that section to make this point even more strongly.